# Chronic Non-Malignant Pain in Patients with Cancer Seen at a Timely Outpatient Palliative Care Clinic

**DOI:** 10.3390/cancers12010214

**Published:** 2020-01-15

**Authors:** David Hui, Eman Abdelghani, Joseph Chen, Shiva Dibaj, Donna Zhukovsky, Rony Dev, Kimberson Tanco, Ali Haider, Ahsan Azhar, Akhila Reddy, Daniel Epner, Joseph Arthur, Shalini Dalal, Yvonne Heung, Suresh Reddy, Maxine De La Cruz, Diane Liu, Eduardo Bruera

**Affiliations:** 1Department of Palliative Care, Rehabilitation and Integrative Medicine, MD Anderson Cancer Center, Houston, TX 77030, USA; EHAbdelghani@mdanderson.org (E.A.); JTChen@mdanderson.org (J.C.); seyedehs@buffalo.edu (S.D.); dzhukovs@mdanderson.org (D.Z.); rdev@mdanderson.org (R.D.); KCTanco@mdanderson.org (K.T.); AHaider@mdanderson.org (A.H.); AAzhar@mdanderson.org (A.A.); ASReddy@mdanderson.org (A.R.); DEpner@mdanderson.org (D.E.); JAArthur@mdanderson.org (J.A.); SDalal@mdanderson.org (S.D.); YJHeung@mdanderson.org (Y.H.); sreddy@mdanderson.org (S.R.); mdelacruz@mdanderson.org (M.D.L.C.); ebruera@mdanderson.org (E.B.); 2Department of Biostatistics, MD Anderson Cancer Center, Houston, TX 77030, USA; dianeliu@mdanderson.org

**Keywords:** analgesics, opioid, cancer-associated pain, chronic pain, neoplasms, palliative care, prevalence, therapeutics

## Abstract

Palliative care is seeing cancer patients earlier in the disease trajectory with a multitude of chronic issues. Chronic non-malignant pain (CNMP) in cancer patients is under-studied. In this prospective study, we examined the prevalence and management of CNMP in cancer patients seen at our supportive care clinic for consultation. We systematically characterized each pain type with the Brief Pain Inventory (BPI) and documented current treatments. The attending physician made the pain diagnoses according to the International Association for the Study of Pain (IASP) task force classification. Among 200 patients (mean age 60 years, 69% metastatic disease, 1-year survival of 77%), the median number of pain diagnosis was 2 (IQR 1–2); 67 (34%, 95% CI 28–41%) had a diagnosis of CNMP; 133 (67%) had cancer-related pain; and 52 (26%) had treatment-related pain. In total, 12/31 (39%) patients with only CNMP and 21/36 (58%) patients with CNMP and other pain diagnoses were on opioids. There was a total of 94 CNMP diagnoses among 67 patients, including 37 (39%) osteoarthritis and 20 (21%) lower back pain; 30 (32%) were treated with opioids. In summary, CNMP was common in the timely palliative care setting and many patients were on opioids. Our findings highlight the need to develop clinical guidelines for CNMP in cancer patients to standardize its management.

## 1. Introduction

Pain is one of the most common and distressing symptoms in cancer patients, occurring in 50% to 90% of patients with advanced malignancies [1,2]. Poorly controlled pain can have a significant impact on patients’ mobility, sleep, mood, function, and overall quality of life [3]. Specialist palliative care teams have significant expertise in pain and symptom management, providing impeccable assessment and diagnosis of pain syndromes, personalized treatment recommendations, and careful monitoring [4]. Under an integrated oncologic-palliative care model, an increased number of cancer patients are being referred to palliative care earlier in the disease trajectory [5]. At MD Anderson Cancer Center, the median survival from the time of outpatient referral to death was greater than 1 year [6]. In these patients, chronic illnesses, such as chronic pain syndromes, can also have a major impact on quality of life.

In patients with cancer, pain can be classified as cancer-related pain, cancer treatment-related pain, or chronic non-malignant pain. There is a paucity of literature on the prevalence and impact of chronic non-malignant pain in the cancer literature. Chronic non-malignant pain has been studied mostly in patients with non-malignant diagnoses while pain research in cancer patients has focused mostly on cancer-related pain and cancer treatment-related pain. Indeed, the Center for Disease Control Clinical Practice Guideline on chronic pain specifically excluded cancer and palliative care patients [7,8,9,10]. At the same time, guidelines on cancer pain have limited discussion on the management of chronic non-malignant pain in cancer patients [11,12].

A systematic and comprehensive examination of the prevalence of chronic non-malignant pain would inform clinical practice in the timely palliative care setting because the management of chronic non-malignant pain can be different from cancer pain. This is a particularly relevant issue in the era of an opioid epidemic, in which multiple pain management guidelines recommend minimizing the prescription of opioids for chronic non-malignant pain (outside of the palliative cancer care setting). In this prospective study, we examined the prevalence and management of chronic non-malignant pain in cancer patients referred to the supportive care clinic at our comprehensive cancer center.

## 2. Results

### 2.1. Patient Characteristics

Recruitment occurred between 16 August 2018 and 14 March 2019. During this period, there were 606 scheduled palliative care consultations. In total, 392 patients were not approached because research staff were not available (*n* = 117), the patient did not show up (*n* = 92), competing studies (*n* = 61), the patient was ineligible (*n* = 45), and other reasons (*n* = 77). Among the remaining 214 patients, 200 (96%) of the 208 fully eligible patients were enrolled.

Table 1 shows the baseline patient characteristics at the supportive care clinic consultation. The mean age was 60 years (range 21, 92), 106 (53%) were female, 153 (77%) were Caucasian, and 141 (71%) were married. The most common types of malignancies were gastrointestinal (*n* = 43, 22%) and respiratory (*n* = 35, 18%), and 138 (69%) had metastatic cancer. Pain was the symptom with the highest intensity on the Edmonton Symptom Assessment System (ESAS) (mean 5.1, SD 2.7). The median number of pain diagnosis was 2 (IQR 1–2) per patient.

### 2.2. Prevalence of Chronic Non-Malignant Pain

Among the 200 patients, 67 (34%, 95% CI 28–41%) had a diagnosis of chronic non-malignant pain; 133 (67%) reported cancer pain, 52 (26%) had treatment-related pain, and 20 (10%) had pain that was not classifiable at the time of assessment. Among the 67 patients with chronic non-malignant pain, 31 (46%) had chronic non-malignant pain only, 20 (30%) had concurrent cancer-related pain, 12 (18%) had concurrent treatment-related pain, and 4 (6%) had both cancer-related pain and cancer treatment-related pain (Figure 1A).

In the univariate analysis (Table 1), patients with chronic non-malignant pain were significantly older (66 vs. 57, *p* < 0.001), less likely to be married (69% vs. 73% *p* = 0.048), more likely to have >1 pain diagnoses (85% vs. 43%, *p* < 0.001), and had a lower morphine equivalent daily dose (MEDD) (median 0 vs. 10, *p* = 0.001). They were also less likely to have metastatic cancer (58% vs. 74%, *p* = 0.06) and had a higher Charlson Comorbidity Index (CCI) (7.2 vs. 6.6, *p* = 0.054), albeit non-statistically significant. Patients with non-malignant pain also had a longer 1-year survival (87% vs. 71%, *p* = 0.03). In the multivariate logistic regression analysis, only the number of pain diagnoses was associated with the presence of chronic non-malignant pain (1 vs. 2: odds ratio 10.7 (95% 2.1–54.6), *p* = 0.71; 1 vs. 3 or more: odds ratio 75.1 (95% CI 10.8, 520.7), *p* < 0.001).

### 2.3. Pain Diagnoses, Characteristics, and Treatments

There was a total of 361 pain diagnoses, with 182 (50%), 60 (17%), 94 (26%), and 25 (7%) classified as cancer-related pain, treatment-related pain, chronic non-malignant pain, and unclassified pain, respectively (Table 2). Physicians rated the level of supporting evidence as definite for 100% of cancer-related pain, 95% of cancer treatment-related pain, and 82% of chronic non-malignant pain (Table 2). Among the 94 chronic pain diagnoses, the most common conditions were osteoarthritis (*n* = 37, 39%), low back pain (*n* = 20, 21%), rheumatoid arthritis (*n* = 5, 5%), and fibromyalgia (*n* = 4, 4%). 

The mean Brief Pain Inventory (BPI) intensity score was 4.0 for cancer-related pain, 3.5 for cancer treatment-related pain, and 3.4 for chronic non-malignant pain (*p* = 0.08), with corresponding mean BPI pain interfere score of 3.5, 2.5, and 2.5, respectively (*p* = 0.004). Among the 67 patients with chronic non-malignant pain, 41 (62%) ranked their chronic non-malignant pain as the most important pain diagnosis (Table 2). The duration of chronic non-malignant pain appeared longer than for the other two pain types (median 17.5 months vs. 3 months, *p* < 0.0001).

### 2.4. Pain Management

When analyzed by patients, 33/67 (49%) patients who reported chronic non-malignant pain were on opioids, in contrast to 97/133 (73%) patients with cancer-related pain and 31/52 (60%) patients with cancer treatment-related pain (Figure 1B). An examination of only rescue opioid use showed similar proportions (Figure 1C).

When analyzed by pain diagnoses, chronic non-malignant pain was most commonly treated with acetaminophen (36%), opioids (32%), and non-steroidal anti-inflammatory drugs (NSAIDs) (29%) (Table 3). For cancer-related pain, the most common therapeutic interventions were opioids (66%), acetaminophen (31%), and non-steroidal anti-inflammatory drugs (NSAIDs) (17%) at the time of palliative care consultation. For cancer treatment-related pain, opioids remained the most commonly used (47%), followed by gabapentinoids (28%) and nothing (23%).

## 3. Discussion

We found that one in three cancer patients seen at the supportive care clinic for timely palliative care consultation had a diagnosis of chronic non-malignant pain. Chronic non-malignant pain often co-existed with other cancer-related pain and cancer treatment-related pain, and half of patients reporting chronic non-malignant pain were using opioids. Our findings highlight the need to develop clinical guidelines specifically for chronic non-malignant pain in cancer patients to standardize its management.

To our knowledge, this prospective study is the first to focus on examining the prevalence of chronic non-malignant pain in palliative cancer care populations. One-third of cancer patients presented with chronic non-malignant pain at consultation. A handful of studies have reported the prevalence of chronic non-malignant pain in cancer patients, which varied between 2% and 76% [13,14]. This wide variation may be explained by differences in the patient population, definitions, and study methodology. Specifically, previous studies were mostly retrospective in nature [14,15,16] and non-malignant pain diagnosis was not the main study outcome [13,17]. Moreover, a majority of studies focused on documenting one main pain site per patient when patients often presented with multiple pain types, which could result in an underestimation. One exception was the study by Grond et al. that assessed 4542 anatomically distinct pain syndromes among 2266 cancer patients referred to an anesthesiology-based pain service in Germany over a 10-year period. They reported that 9% of these patients had pain syndromes unrelated to cancer or its treatment, although 39% of patients had pain for less than one month [18]. 

In contrast to chronic pain in patients without cancer, chronic non-malignant pain in the palliative cancer care setting necessitates special considerations. From a diagnostic standpoint, the identification of cancer-related or treatment-related pain may sometimes be complicated by chronic non-malignant pain syndromes. For instance, spinal cord compression may be masked in a cancer patient with a history of chronic lower back pain. Furthermore, cancer treatments, such as immunotherapy, may exacerbate rheumatoid arthritis and chemotherapy could worsen existing diabetic peripheral neuropathy. Despite these potential challenges, we found that palliative care clinicians were able to make a diagnosis for 93% (336/361) of pain syndromes at the time of consultation and with a high level of supporting evidence.

From a pain management standpoint, chronic non-malignant pain in the cancer patient can pose a challenge clinically. First, cancer patients often have concurrent pain diagnoses and may be prescribed other analgesics for those pain syndromes. For instance, 33 of 67 (49%) cancer patients with chronic non-malignant pain were on opioids, and 17 of 60 (28%) cancer treatment-related pain diagnoses were treated with gabapentinoids. Second, cancer patients have multiple comorbidities, which may either limit pain treatment options (e.g., renal failure and NSAIDs) and/or increase the number of existing medications (i.e., polypharmacy). The mean CCI was 7.2 in this study. Third, chronic pain may significantly limit patients’ function, compounding the impact of cancer on the patient’s quality of life. Fourth, patients with advanced cancer often have a limited prognosis, which may impact their management. Fifth, some patients may be seeing primary care or pain clinics for the management of their chronic non-malignant pain while seeking advice from oncology and/or palliative care for their cancer pain, resulting in a “congress” approach to symptom management [19]. Sixth, concurrent chronic pain syndromes could complicate interpretation of the treatment response because opioids have limited efficacy for chronic non-malignant pain. Given the many unique diagnostic and treatment challenges, further research is needed to assess chronic non-malignant pain in oncology and palliative care patients. 

This topic is of particular relevance because palliative care is seeing a greater number of cancer patients earlier in the disease trajectory [5,20]. Many patients require opioids for symptom management. Proper prescribing of opioids is a critical issue, especially during the opioid epidemic [21]. Unfortunately, there is a lack of pain management guidelines on chronic non-malignant pain in cancer patients because of the dearth of literature on this specific topic. Existing clinical practice guidelines on chronic non-malignant pain specifically exclude cancer and palliative care patients [7,8,9,10]. The few existing guidelines on cancer pain have limited discussion on the treatment of chronic non-malignant pain in this population [11,12,22,23]. 

In the absence of high-quality evidence, how should clinicians manage chronic non-malignant pain in cancer patients? It is important to educate clinicians, patients, and caregivers on the difference between cancer pain and non-malignant pain in terms of natural history and management strategies. Proper treatment strategies of chronic non-malignant pain include rehabilitation therapy, cognitive-behavioral therapy, movement-based interventions, and non-opioid analgesics, such as acetaminophen and non-steroidal anti-inflammatory drugs [7,24]. In contrast to cancer-related pain management, in which opioids are the mainstay, the goal of chronic pain management is to maximize function and minimize harm from long-term pain medication use and misuse. Opioid use for chronic non-malignant pain should be minimized. If opioids are prescribed for cancer-related pain only, patients should be instructed not to take them for chronic non-malignant pain. If patients require opioids for chronic non-malignant pain, risk mitigation strategies, including the use of risk assessment tools, treatment agreements, and urine drug testing, are warranted. Further research is needed to test the different management strategies for chronic non-malignant pain in cancer patients.

This study has several limitations. First, we only enrolled patients from a single comprehensive cancer center. Our findings may not be generalizable to other settings. Patients referred to palliative care generally have a higher symptom burden and more complex pain issues. However, our proportion of patients with chronic non-malignant pain was consistent with the estimate of 10% to 50% in the general population [25]. Second, the number of cancer patients with chronic non-malignant pain was relatively small in this study, which precluded in-depth analysis for this cohort. Future studies may consider target enrollment of these patients. Third, we did not provide longitudinal follow-up to determine pain outcomes in these patients. Further studies are needed to examine longer term outcomes.

## 4. Patients and Methods 

### 4.1. Patients

This was a prospective cross-sectional survey. We recruited patients from the supportivecare outpatient clinic at MD Anderson Cancer Center. Patients were eligible if they were seen for initial consultation, age ≥ 18, had a diagnosis of cancer with or without evidence of active disease, reported having pain within the last 3 months, and did not have delirium (defined as Memorial Delirium Assessment Scale > 13). The Institutional Review Board (2016-0901) at MD Anderson Cancer Center approved this study and all participants provided written informed consent.

### 4.2. Data Collection

We collected patient demographics at enrollment, including Cut down-Annoyed-Guilt-Eye opener-Adapted to Include Drugs (CAGE-AID) questionnaire, the Screener and Opioid Assessment for Patients with Pain-14 (SOAPP14), ESAS, and CCI. The CAGE-AID is a validated questionnaire that examines the pattern of alcohol and drug use. An affirmative answer in at least 2 out of 4 questions has a 70% sensitivity and 85% specificity for substance use disorder [26]. SOAPP14 consists of 14 questions that assess the risk of non-medical opioid use. Each item is rated using a 5-point Likert scale (0 = never, 4 = very often). The total score is 70 and a score > 7 is considered a positive screen [27,28]. The ESAS assesses 10 common symptoms (pain, fatigue, drowsiness, nausea, lack of appetite, dyspnea, depression, anxiety, well-being, and sleep) using an 11-point numeric rating scale (0 = none, 10 = worst) and has been validated in multiple oncology and palliative care settings [29,30]. CCI has 17 questions on various medical conditions, with each assigned a variable weight from 1 to 6 points [31,32]. A higher total score was associated with greater mortality [33].

Our research physician asked patients how many types of pain they had experienced based on duration, location, and previous diagnoses, which were subsequently verified by the outpatient palliative care physician. For each type of pain, we administered the Brief Pain Inventory and documented the current treatment regimen, including opioids. The BPI is a validated questionnaire consisting of 12 numeric rating scales that range from 0 to 10 each (10 = worst) [34]. The pain subscore represents the mean of 4 questions on pain intensity (worst, least, average over past 24 h, and now) and the interference subscore is calculated by averaging 7 questions on activity interference (general activity, mood, walking ability, normal work, relations with other people, sleep, and enjoyment of life).

We asked the attending palliative care physicians to ascertain the clinical diagnosis for each pain type based on the Caraceni and Portenoy classification system, which included 10 major categories and 91 subcategories [35]. The major categories included: (1) Neoplastic damage to bone and joints; (2) neoplastic damage to viscera; (3) neoplastic damage to soft tissue and miscellaneous syndromes; (4) pain syndrome related to direct tumor involvement due to lesions of nervous tissue; (5) intracranial hypertension due to tumor; (6) headache, neck, or back pain due to meningeal disease (does not include radiculopathy); (7) pain syndromes related to therapy; (8) pain associated with cancer-induced complications or debility; (9) unrelated to cancer or its treatment; and (10) unknown. Category 1–7 were considered as cancer-related pain, category 8 was cancer treatment-related pain, and category 9 was considered as chronic non-malignant pain.

The palliative care physician also recorded the supporting evidence for each pain diagnosis based on 4 previously published criteria [36]: (1) Pain with distinct anatomically plausible distribution, (2) a history suggestive of a relevant etiologic factor causing pain (evidence for disease or neural change), (3) demonstration of the pain by at least one confirmatory test, and (4) demonstration of the etiologic factor by at least one confirmatory test. The level of supporting evidence was coded as “possible”, “probable”, and “definite” if the first 2, first 3, and all 4 criteria were fulfilled, respectively [36].

### 4.3. Statistical Analysis

We calculated that 200 evaluable patients would allow us to estimate the proportion of patients who had chronic non-malignant pain with a standard error of 2.5%, assuming that the proportion of chronic non-malignant pain was 15%.

Baseline demographics were summarized using descriptive statistics. Because patients often had more than one pain diagnosis, we analyzed the data by patient (with vs. without non-malignant pain) and also by pain diagnosis (cancer-related pain vs. cancer treatment-related pain vs. chronic non-malignant pain). We estimated the proportion of patients who had chronic non-malignant pain with 95% confidence intervals. To compare among patients with and without non-malignant pain, we used the Wilcoxon rank-sum test or Kruskal–Wallis test for continuous variables, the Chi-squared test or Fisher’s exact test for categorical variables, and the log rank test for survival data. Overall survival was calculated from time of study enrollment to last date of follow-up. To compare among the pain diagnoses, we used generalized linear mixed models for both categorical and continuous variables to account for the partially overlapping data structure because many patients had more than 1 pain diagnosis. Variables with *p* < 0.10 were entered into a multivariable logistic regression model to identify factors associated with chronic non-malignant pain. 

The Statistical Analysis System version 9.4 (SAS Institute Inc., Cary, NC, USA) was used for statistical analysis. A *p*-value of <0.05 was considered to be statistically significant. 

## 5. Conclusions

We found that chronic non-malignant pain was common in cancer patients, often co-presented with cancer-related pain, and one in three patients were already using opioids for their chronic non-malignant pain when presenting to palliative care. Given the frequent use of opioids and the growing concern about non-medical opioid use in multiple countries [37,38], careful diagnosis of pain syndromes, longitudinal patient education, and careful monitoring of aberrant behaviors are essential. More research would inform an evidence-based approach to optimize pain management in patients with cancer.

## Figures and Tables

**Figure 1 cancers-12-00214-f001:**
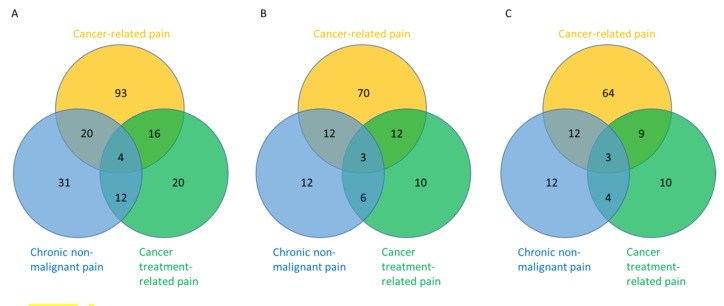
Distribution of pain diagnosis and opioid use. These Venn diagrams illustrate that cancer patients may present with one (non-overlapping) or multiple (overlapping) pain diagnoses at the same time and how opioids were being used. Cancer-related pain is shown in orange, cancer treatment-related pain is shown in green, and chronic non-malignant pain is shown in blue. (**A**) The numbers indicate the count of patients with corresponding pain diagnoses. As shown in the blue circle, a total of 67 (31 + 20 + 4 + 12) patients presented with chronic non-malignant pain, 31 had chronic non-malignant pain alone, and 36 presented with concurrent cancer-related pain and/or cancer treatment-related pain. (**B**) The numbers correspond to the count of patients within each category who were taking opioids (scheduled or rescue). For example, among all 67 patients with chronic non-malignant pain, 33 (12 + 12 + 3 + 6) were on opioids; 12 of 31 (39%) patients who had chronic non-malignant pain without other pain diagnoses were on opioids. (**C**) The use of rescue opioids for chronic non-malignant pain is examined here, with the numbers indicating the count of patients taking rescue opioids for each category. Among the 33 patients with chronic non-malignant pain who were on opioids, 31 (12 + 12 + 3 + 4) were taking them on an as needed basis. That means that only two patients with chronic non-malignant pain (and concurrent cancer treatment related pain) were prescribed scheduled opioid alone without rescue opioids for pain relief. 2.3. Clinical Characteristics associated with Chronic Non-Malignant Pain.

**Table 1 cancers-12-00214-t001:** Baseline characteristics for patients with or without chronic non-malignant pain (*n* = 200).

Variables	Without Chronic Non-Malignant Pain (*n* = 129)	With Chronic Non-Malignant Pain (*n* = 67)	All Patients (*n* = 200) *	*p*-Value
**Age, mean (range)**	57 (21.0, 84.0)	66 (25, 92)	60 (21, 92)	<0.001
**Sex**				
Female	68 (52.7%)	36 (53.7%)	106 (53.0%)	0.89
Male	61 (47.3%)	31 (46.3%)	94 (47.0%)	
**Race/Ethnicity**				
White/Caucasian	96 (74.4%)	54 (80.6%)	153 (76.5%)	0.59
Black/African American	15 (11.6%)	4 (6.0%)	20 (10.0%)	
Asian	4 (3.1%)	1 (1.5%)	5 (2.5%)	
American Indian/Alaskan Native	1 (0.8%)		1 (0.5%)	
Hispanic/Latino	13 (10.1%)	8 (11.9%)	21 (10.5%)	
**Education**				
Some high school or less	4 (3.1%)	4 (6.3%)	8 (4.1%)	0.48
Completed high school	23 (18.1%)	13 (20.3%)	36 (18.6%)	
Some college	33 (26.0%)	19 (29.7%)	52 (26.8%)	
University college	34 (26.8%)	18 (28.1%)	54 (27.8%)	
Advanced degree	33 (26.0%)	10 (15.6%)	44 (22.7%)	
**Marital Status**				
Married	94 (72.9%)	46 (68.7%)	141 (70.5%)	0.048
Single	14 (10.9%)	4 (6.0%)	19 (9.5%)	
Divorced	17 (13.2%)	8 (11.9%)	27 (13.5%)	
Widowed	4 (3.1%)	9 (13.4%)	13 (6.5%)	
**Cancer Diagnosis**				
Head and neck	18 (14.0%)	5 (7.5%)	23 (11.5%)	0.42
Breast	14 (10.9%)	10 (14.9%)	26 (13.0%)	
Respiratory	23 (17.8%)	11 (16.4%)	35 (17.5%)	
Gastrointestinal	28 (21.7%)	15 (22.4%)	43 (21.5%)	
Genitourinary	9 (7.0%)	10 (14.9%)	20 (10.0%)	
Gynecological	11 (8.5%)	3 (4.5%)	14 (7.0%)	
Hematologic	8 (6.2%)	4 (6.0%)	12 (6.0%)	
Other	18 (14.0%)	9 (13.4%)	27 (13.5%)	
**Cancer Stage**				
Localized	12 (9.3%)	10 (14.9%)	22 (11.0%)	0.06
Locally advanced	10 (7.8%)	7 (10.4%)	17 (8.5%)	
Metastatic	96 (74.4%)	39 (58.2%)	138 (69.0%)	
Recurrent	6 (4.7%)	2 (3.0%)	9 (4.5%)	
Other	5 (3.9%)	9 (13.4%)	14 (7.0%)	
**CAGE-AID Score**				
<2	114 (89.1%)	61 (91.0%)	178 (89.4%)	0.67
≥2	14 (10.9%)	6 (9.0%)	21 (10.6%)	
**SOAPP Score, mean (SD)**	4.0 (3.6)	3.5 (3.0)	3.8 (3.4)	0.42
**Charlson Comorbidity Index, mean (SD)**	6.6 (2.4)	7.2 (3.1)	6.8 (2.6)	0.054
**Karnofsky Performance Status, mean (SD)**	73.1 (15.7)	76.4 (14.1)	74.2 (15.2)	0.19
**Number of pain types**				
1	73 (56.6%)	10 (14.9%)	84 (42.0%)	<0.001
2	41 (31.8%)	35 (52.2%)	79 (39.5%)	
≥3	15 (11.6%)	22 (32.8%)	37 (18.5%)	
**Morphine equivalent daily dose, mg/day**				
Mean (SD)	18 (40)	10 (13)	16 (35)	0.001
Median (IQR)	5.0 (3.0, 13.0)	5.6 (1.0,17.0)	5.0 (2.8, 15)	
**One-year survival % (95% CI)**	71 (61, 79)	87 (77, 94)	77 (70, 83)	0.03
**Edmonton Symptom Assessment System, mean (SD)**				
Pain	5.1 (2.7)	5 (2.7)	5.1 (2.7)	0.74
Fatigue	4.8 (3)	5 (2.8)	4.9 (2.9)	0.63
Nausea	2 (2.9)	1.1 (2.1)	1.7 (2.7)	0.01
Depression	1.7 (2.4)	1.4 (2.4)	1.6 (2.4)	0.06
Anxiety	2.4 (2.7)	2.1 (2.9)	2.3 (2.7)	0.17
Drowsiness	3.2 (3)	2.4 (2.8)	2.9 (2.9)	0.06
Dyspnea	2.2 (2.8)	1.7 (2.3)	2 (2.7)	0.31
Appetite	4.2 (3)	3.4 (2.6)	4 (2.9)	0.07
Well-being	4.6 (2.6)	4.1 (2.2)	4.4 (2.5)	0.17
Sleep	4.4 (2.8)	4.4 (2.6)	4.4 (2.7)	0.96
Financial distress	2.2 (2.7)	2 (2.7)	2.1 (2.7)	0.51
Spiritual pain	1 (2)	0.6 (1.3)	0.9 (1.8)	0.37

Abbreviations: CAGE-AID, Cut down-Annoyed-Guilty-Eye opener-Adapted to Include Drugs questionnaire; IQR, interquartile range; SD, standard deviation; SOAPP14; Screener and Opioid Assessment for Patients with Pain-14; * This study enrolled 200 patients, including 67 patients with chronic non-malignant pain, 129 patients without chronic non-malignant pain, and 4 patients with pain syndrome that could not be classified due to inadequate clinical information.

**Table 2 cancers-12-00214-t002:** Characteristics of pain diagnoses (*n* = 361).

Variables	Cancer-Related Pain (*n* = 182)	Cancer Treatment-Related Pain (*n* = 60)	Non-Malignant Pain (*n* = 94)	Total (*n* = 361) *	*p*-Value ǂ
**Pain rank**					
1 (most important)	115 (63.2)	32 (53.3)	41 (43.6)	200 (55.4)	0.003
2	51 (28.0)	17 (28.3)	34 (36.2)	115 (31.9)	
≥3	16 (8.8)	11 (18.3)	19 (20.2)	46 (12.7)	
**Duration in months** median (IQR)	3.0 (1.5, 8.0)	3.0 (1.0, 12.0)	17.5 (3.0, 96.0)	4.0 (1.5, 12.0)	<0.0001
**Brief Pain Inventory**					
Worst pain intensity	6.1 (2.9)	5.4 (2.6)	5.4 (2.7)	5.8 (2.8)	0.01
Least pain intensity	2 (1.9)	2 (1.9)	1.8 (1.7)	1.9 (1.9)	0.64
Average pain intensity	4.2 (2.5)	3.5 (2.3)	3.7 (2.2)	3.9 (2.4)	0.08
Pain intensity now	3.6 (2.8)	3 (2.3)	2.7 (2.4)	3.2 (2.6)	0.03
Pain relief %	5.6 (3.1)	5.6 (3.2)	4.9 (3.2)	5.4 (3.2)	0.41
Activity interference	4.7 (3.5)	3.2 (3.3)	3.5 (3.1)	4.1 (3.4)	0.006
Mood interference	3.2 (3.2)	2.5 (3.3)	1.9 (2.8)	2.7 (3.1)	0.006
Walking interference	2.8 (3.4)	2.4 (3.4)	3.1 (3.4)	2.8 (3.4)	0.52
Work interference	4.3 (3.6)	2.3 (3.2)	3.5 (3.4)	3.7 (3.5)	0.002
Relations interference	1.8 (2.7)	1.3 (2.5)	0.9 (2.3)	1.4 (2.6)	0.03
Sleep interference	4.2 (3.7)	3.5 (3.4)	2.8 (3.4)	3.7 (3.6)	0.02
Enjoyment interference	3.5 (3.3)	2.4 (2.9)	2 (3)	2.9 (3.2)	0.003
BPI pain intensity score	4.0 (2.2)	3.5 (2.0)	3.4 (1.9)	3.7 (2.1)	0.08
BPI pain interference score	3.5 (2.5)	2.5 (2.4)	2.5 (2.3)	3.0 (2.5)	0.004
**Personalized pain goal** mean (SD)	2.2 (1.1)	2.1 (1)	2.1 (1.3)	2.2 (1.1)	0.64
**Level of supporting evidence**					
Definite	180 (100.0)	57 (95.0)	77 (81.9)	314 (94.0)	NA
Possible	0 (0)	1 (1.7)	7 (7.4)	8 (2.4)	
Probable	0 (0)	2 (3.3)	10 (10.6)	12 (3.6)	

Abbreviations: NA, not available; * This study included a total of 361 pain diagnoses, including 67 patients with chronic non-malignant pain, 129 patients without chronic non-malignant pain, and 4 patients with pain syndrome that could not be classified due to inadequate clinical information. Thus, the sum of “cancer-related pain”, “cancer treatment-related pain” and “non-malignant pain” did not always add up to the last column. ǂ Generalized linear mixed model was used to compare the variables among the 3 pain diagnoses categories. Statistical testing was not conducted for some variables because of the very small numbers or proportions in some cells (marked as “NA”).

**Table 3 cancers-12-00214-t003:** Treatment for pain diagnoses (*n* = 361).

Variables	Cancer-Related Pain (*n* = 182)	Cancer Treatment-Related Pain (*n* = 60)	Non-Malignant Pain (*n* = 94)	Total (*n* = 361) *	*p*-Value ǂ
**Treatment at baseline prior to palliative care consultation**					
Acetaminophen	56 (30.8)	10 (16.7)	34 (36.2)	108 (29.9)	0.045
NSAIDs	31 (17.0)	10 (16.7)	27 (28.7)	72 (19.9)	0.26
Opioids	120 (65.9)	28 (46.7)	30 (31.9)	188 (52.1)	<0.0001
Gabapentinoids	18 (9.9)	17 (28.3)	3 (3.2)	39 (10.8)	0.0001
Tricyclic antidepressants	0 (0)	0 (0)	0 (0)	0 (0)	NA
**Serotonin and norepinephrine reuptake inhibitors**	0 (0)	0 (0)	0 (0)	0 (0)	NA
Corticosteroids	3 (1.6)	0 (0)	0 (0)	4 (1.1)	NA
Psychotherapy	0 (0)	0 (0)	0 (0)	0 (0)	NA
Physical therapy	2 (1.1)	0 (0)	2 (2.1)	4 (1.1)	NA
Nothing	28 (15.4)	14 (23.3)	20 (21.3)	69 (19.1)	0.36
Other	8 (4.4)	6 (10.0)	8 (8.5)	24 (6.6)	0.33
**Proportion of rescue opioids for pain diagnosis**					
0%	73 (40.1)	37 (61.7)	65 (69.1)	190 (52.6)	NA
1–10%	4 (2.2)	1 (1.7)	0	5 (1.4)	
11–30%	8 (4.4)	1 (1.7)	6 (6.4)	17 (4.7)	
31–50%	54 (29.7)	7 (11.7)	16 (17.0)	84 (23.3)	
51–99%	1 (0.5)	0	0	1 (0.3)	
100%	42 (23.1)	14 (23.3)	7 (7.4)	64 (17.7)	

Abbreviations: NA, not available. * This study included a total of 361 pain diagnoses, including 67 patients with chronic non-malignant pain, 129 patients without chronic non-malignant pain, and 4 patients with pain syndrome that could not be classified due to inadequate clinical information. Thus, the sum of “cancer-related pain”, “cancer treatment-related pain” and “non-malignant pain” did not always add up to the last column. ǂ Generalized linear mixed model was used to compare the variables among the 3 pain diagnoses categories. Statistical testing was not conducted for some variables because of the very small numbers or proportions in some cells (marked as “NA”).

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
