# Peer review of "Chronic Non-Malignant Pain in Patients with Cancer Seen at a Timely Outpatient Palliative Care Clinic"

_cancers, 2020, doi:10.3390/cancers12010214_

Round 1
Reviewer 1 Report
Dear Authors
Thank you for addressing quite important issue in everyday Palliative Care clinical practice, regarding other than cancer related chronic pains.
Whatsoever, in my opinion, study has got some very important limitations, which concluded me to reccommend the major revision of article before potential resubmitting.
The first and the most important problem is the statistics. Among the 200 patients included to the study only 31 (sic!, =15,5 %) had chronic non-malignant pain without cancer related pain. In my opinion this is a group that should be implemented to the statistics, not other groups. So if you could possibly find more of the similar patients, it would be appropriate to include them to the study and derive conclusions . As well it should be taken into consideration of the cancer itself influence on patients' self-reports of pain (it is well known, that perception of the cancer disease is changing the perception of pain, even when it is not an cancer related pain. So it should be known the patients reports of chronic non-malignant pain before the cancer diagnose and after. The problem of different types of pains in the same patient is not a new one. Almost all recommendations (if not all) are outlining the existence of other than cancer related pains. They are not focusing on it, which is obvious- the cancer related pain will be of the most importance and the multimodal therapy wiil be set on chronic cancer pain (but in the same time, this therapy, and of course, diagnostics, will cover also non-malignant one). The opoid analgesics are still the mainstream of therapy for moderate to severe cancer pain, and no other type of drugs or procedure is going to change them in the near future. I do not think we should produce a specific recommendations or guidelines for patients with chronic malignant and subsequent non-malignant pain There are a lot of guidelines (IASP, EFIC, ASCO etc.)for treating the pain in cancer survivors, where the focus on different treatments and caution to opioid use is strongly emphasized. Last but not least I can not agree about conclusion of opid epidemic in many countries. It is mostly US problem, surely not a European (problems of opioidophobia rather)
Concluding- the study would be of value when the pure group of cancer patients with chronic non-malignant pain only will be recruited.
Author Response
The first and the most important problem is the statistics. Among the 200 patients included to the study only 31 (sic!, =15,5 %) had chronic non-malignant pain without cancer related pain. In my opinion this is a group that should be implemented to the statistics, not other groups. So if you could possibly find more of the similar patients, it would be appropriate to include them to the study and derive conclusions .
REPLY: Thank you for your comments. The primary objective of this study was to examine the prevalence of chronic non-malignant pain in cancer patients with or without co-existing caner pain (n=67, 34%). Because of this objective, it was important to not only focus on those patients, but to enroll other patients without chronic non-malignant pain to understand the denominator. Our sample size of 200 patients was specifically calculated to address this objective. Based on reviewer’s comment, we have now stated in the limitations: “the number of cancer patients with chronic non-malignant pain was relatively small in this study, which precluded in-depth analysis for this cohort. Future studies may consider target enrollment of these patients.”
As well it should be taken into consideration of the cancer itself influence on patients' self-reports of pain (it is well known, that perception of the cancer disease is changing the perception of pain, even when it is not an cancer related pain. So it should be known the patients reports of chronic non-malignant pain before the cancer diagnose and after. The problem of different types of pains in the same patient is not a new one. Almost all recommendations (if not all) are outlining the existence of other than cancer related pains. They are not focusing on it, which is obvious- the cancer related pain will be of the most importance and the multimodal therapy wiil be set on chronic cancer pain (but in the same time, this therapy, and of course, diagnostics, will cover also non-malignant one).
REPLY: Based on the reviewer’s comments, we have now stated in results (section 2.4) that “The mean BPI intensity score was 4.0 for cancer-related pain, 3.5 for cancer treatment-related pain and 3.4 for chronic non-malignant pain (P=0.08), with corresponding mean BPI pain interfere score of 3.5, 2.5 and 2.5, respectively (P=0.004). Among the 67 patients with chronic non-malignant pain, 41 (62%) ranked their chronic non-malignant pain as the most important pain diagnosis (Table 2).” This suggests that while cancer pain was important, chronic non-malignant pain also had significant clinical implications.
The opoid analgesics are still the mainstream of therapy for moderate to severe cancer pain, and no other type of drugs or procedure is going to change them in the near future. I do not think we should produce a specific recommendations or guidelines for patients with chronic malignant and subsequent non-malignant pain There are a lot of guidelines (IASP, EFIC, ASCO etc.)for treating the pain in cancer survivors, where the focus on different treatments and caution to opioid use is strongly emphasized.
REPLY: Based on your feedback, we have reviewed these guidelines again in further detail (IASP, EFIC, ASCO, and ESMO). To our knowledge, these guidelines did not make a clear distinction between malignant and non-malignant pain in cancer patients when providing treatment recommendations. We have now included these references in the discussion.
Last but not least I can not agree about conclusion of opid epidemic in many countries. It is mostly US problem, surely not a European (problems of opioidophobia rather
REPLY: Thank you. We have now revised this statement in the conclusion to “Given the frequent use of opioids and the growing concern about non-medical opioid use in multiple countries, careful diagnosis of pain syndromes, longitudinal patient education and careful monitoring of aberrant behaviours are essential.” We have also provide 2 supporting references.
Concluding- the study would be of value when the pure group of cancer patients with chronic non-malignant pain only will be recruited.
REPLY: Thank you. See response to first question.
Reviewer 2 Report
Thank you very much for your work. I think that is a interesting topic, but we advised to I proved the methodology
You must include in the discussion the cancer treatment related pain, in this work only 28% are adequately treated
In the other hand, it would be important to analyze the adequacy of opioid treatment in non malignant pain into account life expectancy
Author Response
You must include in the discussion the cancer treatment related pain, in this work only 28% are adequately treated
REPLY: We appreciate your thoughtful feedback. We have now added in discussion “17 of 60 (28%) cancer treatment-related pain diagnoses were treated with gabapentinoids”.
Because this was a cross sectional study with patients seeing palliative care for the first time and only a one time assessment, it was difficult to know if the pain was adequately treated. We have now stated in the limitations paragraph of conclusion that “Third, we did not provide longitudinal followup to determine pain outcomes in these patients. Further studies are needed to examine longer term outcomes.”
In the other hand, it would be important to analyze the adequacy of opioid treatment in non malignant pain into account life expectancy
REPLY: Thank you. We have now stated in the limitations paragraph that “Third, we did not provide longitudinal followup to determine pain outcomes in these patients. Further studies are needed to examine longer term outcomes.”
Round 2
Reviewer 1 Report
Dear Authors
Thank you for reply and some explanations you made. An updated article sounds much better to me.
I am agree, fully agree, that we as a pain/palliative care physicians should pay an speciall focus on pain (not only cancer-related but also caused by co-morbidities). But we should also be very carefull with producing more and more detailed guidelines with low evidence, while the existing ones are not fully known by most of the physicians.
I agree taht non-cancer pain problem is not clearly distinguished in many guidelines, but don't you think that having guidelines on cancer- and non-cancer pain, the diagnose and assessment is of importance (see below:"Standards for the management of cancer-related pain across Europe. A position paper from EFIC Task Force on Cancer Pain". Bennett M.I. et al.; Eur J Pain, 23 (4), 660-668, 2019)
2.2 Standard 2. Patients identified with cancer‐related pain should receive a pain assessment when seen by a healthcare professional, which at a minimum classifies the cause of pain based on proposed ICD‐11 taxonomy and establishes the intensity and impact on quality of life of any pain that they report. [GRADE 1B]
Cancer‐related pain has multiple aetiologies including the cancer itself (cancer pain) and cancer treatments, particularly surgery, chemotherapy (including hormonal, biological and immune therapies) and radiotherapy (Bennett, 2017). It can originate from visceral, bone or nerve tissues and can have nociceptive, neuropathic or inflammatory mechanisms (Falk and Dickenson, 2014; Knudsen et al, 2009). It also varies in its temporal characteristics: it can be acute or chronic and may have continuous or episodic features (Caraceni and Portenoy, 1999). Persistent cancer pain can lead in some individuals to the development of chronic widespread pain induced by plastic changes in the somatosensory nervous system) Kosek et al, 2016).
Pain may also be caused by comorbid conditions unrelated to cancer, and this aetiology accounts for around 10%–20% of pains in cancer patients (Bennett et al., 2012; Grond et al., 1996). Therefore, pain in a cancer patient is not synonymous with cancer‐related pain; a clinical assessment must distinguish between cancer pain, cancer treatment pain and pain from comorbid conditions.
Often, patients experience mixed types of pain simultaneously, or pain that changes over time. A detailed diagnostic assessment that must be repeated at appropriate intervals is therefore often required to guide treatment strategies.
A bedside assessment can determine the intensity, aetiology, character and underlying mechanisms of pain leading to improvements in pain outcomes (Trowbridge et al, 1997). The new proposed ICD‐11 classification for cancer‐related pain (Table 1) will enable a standardized taxonomy for clinical practice and research and should be used widely (Bennett et al., 2019; Treede et al, 2015). The Brief Pain Inventory provides a summary of the severity and impact of the pain on the patient’s daily activities (Cleeland and Ryan, 1994). Additionally, conditions which can amplify pain expression such as distress, anxiety and depression (Laird et al, 2016), delirium or effects of alcohol and mis‐used drugs, should be incorporated in a structured pain assessment (Table 2).
I hope your article will help to open a discussion not only on pain diagnose and treatment, but also on education.
Author Response
1. I agree taht non-cancer pain problem is not clearly distinguished in many guidelines, but don't you think that having guidelines on cancer- and non-cancer pain, the diagnose and assessment is of importance (see below:"Standards for the management of cancer-related pain across Europe. A position paper from EFIC Task Force on Cancer Pain". Bennett M.I. et al.; Eur J Pain, 23 (4), 660-668, 2019)
REPLY: Although some guidelines suggested we should assess for non-cancer causes of pain in cancer patients, they did not state how to treat non-malignant pain in cancer patients (In discussion "The few existing guidelines on cancer pain have limited discussion on the treatment of chronic non-malignant pain in this population [11, 12, 22, 23]."). Thus, we advocate for further research (In discussion: "Given the many unique diagnostic and treatment challenges, further research is needed to assess chronic non-malignant pain in oncology and palliative care patients.") and discussion on this topic and we believe this manuscript would inform this discussion. Thank you.
2. I hope your article will help to open a discussion not only on pain diagnose and treatment, but also on education.
REPLY: We agree with education and have now added more on this in the discussion: "It is important to educate clinicians, patients and caregivers the difference between cancer pain and non-malignant pain in terms of natural history and management strategies. "
Reviewer 2 Report
The paper is similar to previos version. In my opinion this paper don´t contribute anything new; the analysis and the discussion are poor.
In addition the material and methods are still located after the discussion
Author Response
The paper is similar to previos version. In my opinion this paper don´t contribute anything new; the analysis and the discussion are poor.
REPLY: Thank you for your comments. We have tried to be as specific as possible when responding to the reviewer's comments. We will ask the editor to make the final decision regarding this manuscript.
In addition the material and methods are still located after the discussion
REPLY: This is the format of this journal. Thank you.